# Complex Valued Risk Diversification

**DOI:** 10.3390/e21020119

**Published:** 2019-01-28

**Authors:** Yusuke Uchiyama, Takanori Kadoya, Kei Nakagawa

**Affiliations:** 1MAZIN, Inc., 1-60-20 Minami-Otsuka, Toshima-ku, Tokyo 170-0005, Japan; 2Nomura Asset Management Co., Ltd. 1-21-1, Nihonbashi, Chuo-ku, Tokyo 103-8260, Japan

**Keywords:** portfolio management, risk diversification, Hilbert transform, principal component analysis

## Abstract

Risk diversification is one of the dominant concerns for portfolio managers. Various portfolio constructions have been proposed to minimize the risk of the portfolio under some constraints, including expected returns. We propose a portfolio construction method that incorporates the complex valued principal component analysis into the risk diversification portfolio construction. The proposed method was verified to outperform the conventional risk parity and risk diversification portfolio constructions.

## 1. Introduction

Both individual and institutional investors are concerned with risk diversification for portfolio construction. Portfolio managers have employed appropriate mathematical techniques to minimize the risk of the portfolios, formulated as constrained nonlinear optimization problems. Indeed, as the pioneer of quantitative finance, Markowitz proposed the mean-variance (MV) portfolio construction [1]. The heart of the MV portfolio is “don’t put all your eggs in one basket”. We can reduce the risk of the portfolio by combining different assets together which are not highly correlated. However, it has been pointed out that the asset allocations of the MV portfolio are often biased [2]. In other words, the weight levels of particular assets are much higher than others in the MV portfolio.

In general, risk biased portfolios seem to be vulnerable to asset price changes. The MV portfolio construction is thus undesirable from the point of view of risk diversification. The risk parity (RP) portfolio construction was designed to allocate market risk equally across asset classes, including stocks, bonds, commodities, and so on [3]. Subsequently, a return weighted sum of the assets is introduced to the RP portfolio construction to improve its performance [4]. Some variations of the RP portfolio construction have been proposed and verified to outperform the MV portfolio construction [5,6,7]. Nevertheless, the RP portfolio construction cannot fully disperse the origin of risk because almost all parts of the world mutually interact in modern society, causing entanglement of different asset classes.

In the field of data science and multivariate analysis, principal component analysis (PCA) has been developed to decompose mutually correlated data subspaces [8]. The maximum risk diversification (MRD) portfolio construction uses the PCA to decompose and allocate the risk contribution of assets [9]. Then. the constrained optimization of the MRD portfolio construction is expected to diversify the origin of risks. The MRD portfolio is also confirmed to outperform the MV portfolio and to be able to allocate the risk contribution of assets [9,10]. After the concept and method of the MRD portfolio was proposed, this method was tested and has been used in both academia and industry. Recently, in fact, the availability of the MRD portfolio construction was reported by an empirical test for commodities [11].

On the other hand, in the field of the atmospheric physics, the PCA has been used and extended to capture principal modes of spatiotemporal dynamics, which are known as empirical orthogonal functions (EOFs) [12]. In practice, it is extremely difficult to investigate all the degrees of freedom of global atmospheric changes. Thus, the method of EOFs has been employed to extract essential dynamics [13,14,15].

Conventional portfolio constructions are based on empirical covariance matrices, which are inferred by the returns of assets. Conventional portfolio constructions with empirical cpvariance matrices, however, cannot incorporate the dynamic properties of the assets into the portfolios. In this research, we introduced the method of EOFs to the MRD with the aid of the Hilbert transform. In the area of signal processing and machine-learning, the Hilbert transform (with empirical mode decomposition) was used to capture the dynamic property of time-series and to generate feature variables for prediction models [16]. It is thus expected that the proposed portfolio construction will outperform conventional portfolio constructions by including the dynamic properties of the risks, which are derived from the Hilbert transform of the returns of the assets.

## 2. Related Works

This section reviews the conventional portfolio constructions, namely, the MV, RP, and MRD portfolio constructions. Table 1 shows a summary of the conventional portfolios in terms of “What is diversified?” and “Risk measure”. Specifications of them are shown in the following subsections.

### 2.1. Mean-Variance Portfolio

Markowitz first introduced the MV portfolio construction as a sophisticated method in modern portfolio theory. In this theory, the risk of an asset is defined as the standard deviation of the return. With this setup, a portfolio is designed by the weighted sum of the assets considered.

Given the sequence of *m*-th asset prices ${\left\{p_{t}^{\left(m\right)}\right\}}_{0\leq t\leq T}\;\left(1\leq m\leq M\right)$, the return of the asset is defined by:
(1)$$r_{t}^{\left(m\right)}=\frac{p_{t+1}^{\left(m\right)}-p_{t}^{\left(m\right)}}{p_{t}^{\left(m\right)}}.$$


Subsequently, the return of the portfolio is obtained as:
(2)$$R_{t}=\sum_{m=1}^{M}w_{m}r_{t}^{\left(m\right)},$$
where ${\left\{w_{m}\right\}}_{1\leq m\leq M}$ is the set of weight coefficients of the portfolio. The risk of the portfolio is defined by the standard deviation of the return in Equation (Equation 2). In general, risk-averse investors tend to minimize the risk of their portfolios under expected returns. This strategy is mathematically formalized by constrained quadratic programming with respect to the covariance matrix of the return of the portfolio.

The expected return of the portfolio in Equation (Equation 2) is expressed by the weighted sum of the expected return of each asset as:
(3)$$E\left[R_{t}\right]=\sum_{m=1}^{M}w_{m}E\left[r_{t}^{\left(m\right)}\right],$$
where E[·] denotes the expectation for a random variable. The covariance matrix of the return of the portfolio is defined by:
(4)$$\Sigma =E\left[\left(r_{t}-E\left[r_{t}\right]\right){\left(r_{t}-E\left[r_{t}\right]\right)}^{T}\right],$$
where components of $r_{t}$ are the return of each asset and ${\left(\cdot \right)}^{T}$ denotes the transpose of a vector. With the use of the covariance matrix in Equation (Equation 4), the variance of the portfolio is obtained as:
(5)$$\sigma^{2}=w^{T}\Sigma w,$$
with w being a weight coefficient vector. The MV optimized portfolio with expected return μ is realized as the solution of the minimization for $\sigma^{2}$ in Equation (Equation 5) subject to $w^{T}r_{t}=\mu$. Additionally, constraints for the weight coefficients can be added to the objective function as a Lagrangian form with multipliers.

### 2.2. Risk Parity Portfolio

It has been pointed out that the asset classes of the MV portfolio are not fully allocated. To disperse the risk contributions of portfolios, risk parity (RP) portfolio constructions have been proposed. Based on the idea of the RP portfolio construction, a measure of risk contribution was introduced.

The risk contribution of the *m*-th asset is derived from the variance of the RP portfolio as follows:
(6)σm=wm∂σ∂wm(7)=(Σw)mwTΣw,
where ${\left(\Sigma w\right)}_{m}$ denotes the *m*-th component of Σw. Equal risk contribution for the RP portfolio requires that all the risk contributions have the same value, whereby the weight coefficients of the portfolio are determined by optimization as follows:
(8)$$\underset{w}{\arg\;\min}\sum_{m=1}^{M}\left[w_{m}-\frac{\sigma }{{\left(\Sigma w\right)}_{m}M}\right].$$


This portfolio construction enables one to obtain equally allocated assets. In addition, various subclasses of the RP portfolio construction have been proposed. For instance, the return weighted RP portfolio construction was developed to improve the performance of the equally risk-allocated RP portfolio [4].

### 2.3. Risk Diversification

In general, the origin of the risk of assets seems to be entangled. Namely, the covariance matrix of the return of portfolios contains nondiagonal components, and thus, the pair of assets exhibit a linear correlation. To unravel the entangled risks, PCA has been incorporated into the portfolio constructions [9].

The covariance matrix of the return of portfolios can be transformed into a diagonal matrix by an appropriate orthogonal matrix, since all of the eigenspaces are mutually independent. The eigenvalues of the covariance matrix introduce a probability distribution of risk contribution. Thus, the entropy with respect to the probability distribution is defined and is employed as the objective function of the MRD portfolio construction. As is well known, the principle of maximum entropy introduces the most diversified probability distribution under given constraints. In this case, such a probability distribution of risk contributions is desirable to allocate the origin of risks related to the assets of the portfolios.

## 3. Complex Valued Risk Diversification Portfolio Construction

As has been reviewed in the previous section, almost all of the portfolio construction methods use the covariance matrix to estimate the risk of the portfolios as the objective functions of the optimization. The covariance matrix of a random vector contains autocorrelations of pairs of vector components. Thus, one can extract stationary information of the random vector from the corresponding covariance matrix. However, in general, the price of an asset exhibits nonstationary random fluctuations. Hence, it is necessary to use dynamic information of the fluctuations of the assets to accurately estimate the risk of the portfolios.

In order to incorporate the dynamics of the price of the assets into the portfolio constructions, we applied the method of EOFs to the time-series of the return of the assets. This portfolio construction method uses a complex valued time-series derived from the Hilbert transform and the corresponding covariance matrix, which we name a complex valued risk diversification (CVRD) portfolio construction.

As well as the MRD portfolio construction, the CVRD portfolio construction uses the PCA to reduce the dimension of the data space. In general, real financial data might have nonlinear correlations. Hence, nonlinear dimensionality reductions, such as Isomap [17] and kernel PCA [18], would be applied for accuracy and extract dominant subspaces. However, weights of the dominant subspaces, which are derived by the nonlinear dimensionality reductions, do not correspond to those of assets in the portfolios because nonlinear maps mix the linear dependency of the assets. Thus, we employ the PCA, despite its linearity, as the dimensionality reductions in order to estimate the weights of the assets in the portfolios directly.

The Hilbert transform of a time-series x(t) on t∈[0,∞) is defined by:
(9)$$H\left[x\left(t\right)\right]=\frac{1}{\pi }\int_{0}^{\infty }\frac{x(\tau )}{t-\tau }d\tau ,$$
where the improper integral is understood in the sense of Cauchiy’s principal value [19]. In practice, empirical time-series are recoded at a certain sampling rate Δt, which introduces discrete time $t_{n}=n\Delta t$ with *n* being integer. The Hilbert transform for a discrete time-series is given by:
(10)$$H_{D}\left[x_{k}\right]=-i\;\mathrm{sgn}\left(k-\frac{N}{2}\right)\sum_{n=0}^{N-1}x_{n}e^{i\frac{2\pi n}{N}},$$
where sgn(·) is the sign function [20]. Here, we apply the Hilbert transform in Equation (Equation 10) to the return of the portfolio in Equation (Equation 2) and then obtain the analytic signal as:
(11)$$z_{t}=r_{t}+iH_{D}\left[r_{t}\right].$$


As well as the PCA for real valued time-series, the analytic signal, $z_{t}\;\left(0\leq t\leq T\right)$, provides a complex valued covariance matrix defined as:
(12)$$C_{z}=E\left[\left(z_{t}-E\left[z_{t}\right]\right){\left(z_{t}-E\left[z_{t}\right]\right)}^{*}\right],$$
with $z_{t}^{*}$ being the transjugate of $z_{t}$. Since $C_{z}$ in Equation (Equation 12) is a positive definite Hermitian matrix, the corresponding eigenvalues are positive real values including zeros. A unitary matrix *U*, which consists of eigenvectors of $C_{z}$, transforms $C_{z}$ into:
(13)$$UC_{z}U^{*}=\Lambda ,$$
where Λ is the orthogonal matrix which consists of eigenvalues $\left\{\lambda_{n}\right\}$ of $C_{z}$. The weight coefficient vector w, at the same time, is transformed into $\tilde{w}=Uw$. The indicator of contribution of the eigenvector is introduced as:
(14)$$v_{m}={\tilde{w}}_{m}^{2}\lambda_{m},$$
with ${\tilde{w}}_{m}$ being the *m*-th component of the transformed weight coefficient $\tilde{w}$, and the probability distribution for $v_{m}$ is defined by:
(15)$$p_{m}=\frac{v_{m}}{\sum_{m=1}^{M}v_{m}}.$$


From the probability distribution in Equation (Equation 15), the corresponding entropy can be introduced as:
(16)$$H=-\sum_{m=1}^{M}p_{m}\log p_{m}.$$


In general, weight coefficients of portfolios are constrained based on trading strategies. Thus, we constructed a Lagrangian function with the entropy in Equation (Equation 16) and constraint functions for weight coefficients as:
(17)$$L=H-\sum_{l=1}^{L}\mu_{l}g_{l}\left(\tilde{w}\right),$$
where $\mu_{l}$ is a Lagrange multiplier and $g_{l}\left(\cdot \right)$ is a constraint function. Optimizing *L* in Equation (Equation 17) with respect to w gives the weight coefficients of the expected CVRD portfolio.

## 4. Empirical Analysis

In this section, we test the performance of the CVRD portfolio construction proposed in Section 3 by comparing with the conventional portfolio constructions reviewed in Section 2, the RP and MRD portfolio constructions. Data description and the results of performance test are given in the following subsections.

### 4.1. Data Description

To evaluate the performance of each portfolio, we selected six commodities, six indices, and five global currencies to US dollar, as shown in Table 2. All of the daily historical data were corrected during May 2000 to April 2017 and transformed into returns to estimate the covariance matrices. The descriptive statistics of the returns of the assets are shown in Table 3.

### 4.2. Performance Test of Portfolios

In order to implement portfolio optimizations, we used a real valued empirical covariance matrix for the RP and MRD portfolio constructions and a complex valued one for the CVRD portfolio construction. It has been pointed out, by the random matrix theory, that the empirical covariance matrix cannot infer the true value of a covariance matrix [21,22,23]. Nevertheless, the empirical covariance matrix has been used to construct portfolios in industry because of ease of estimation for practitioners. Based on the correction formula proposed in Reference [23], the empirical return of the portfolio is $1/\sqrt{1-p/n}$ times larger than its true value, where *p* is number of assets in the portfolio and *n* is the length of historical data. In this case, p=17 and n=252, then the correction ratio is evaluated as $1/\sqrt{1-p/n}\approx 1.034$. Thus, we think that the empirical covariance matrix can be used as a better inference of the true covariance matrix.

The optimizations of the RP and MRD portfolio constructions were implemented by the presented method in Section 2 with the empirical covariance matrix. To evaluate the complex valued covariance matrix, we employed the discrete Hilbert transform for the daily returns. Then, PCA was executed to the real and complex valued covariance matrices with the aid of the Jacobi method. Newton’s method was employed for the optimization of each portfolio construction. The weights of the all of the portfolios were rebalanced every year without transaction costs. As performance measures of the portfolios, we employed averaged return and risk, and Sharpe ratio, which are often used in industry. In addition, we introduced omega ratio because it can include all of the statistical information of the portfolios [24].

Table 4 shows the annual return, risk, Sharpe ratio, and omega ratio of the RP, MRD, and CVRD portfolios. The CVRD portfolio outperforms the MRD with respect to all of the measures. The return and omega ratio of the CVRD portfolio is better than those of the RP. On the other hand, the risk and Sharpe ratio of the RP portfolio is better than those of others. Table 5 and Table 6 show the time averaged weight coefficients of the portfolios in terms of assets and type of assets, respectively. The corresponding sequences of the weights of the RP and CVRD portfolios are shown in Figure 1 and Figure 2. It is seen that the RP portfolio mainly consists of the commodities whereby the risk of it is the lowest in the portfolios. On the other hand, the CVRD portfolio has the indices much larger than the RP and MRD portfolios. Hence, the return of the CVRD outperforms other portfolios. In fact, as is shown in Figure 3, the cumulative return of the CVRD is on a higher level than that of the RP and MRD portfolios.

Note that, as well as the MRD portfolio construction, the rebalancing frequency of the CVRD portfolio construction tends to be higher than the conventional ones. such as the MV and RP portfolio constructions, while the origins of risks can be diversified suitably. Additionally, the risk levels of the assets are not considered in the processes of the portfolio optimizations. Thus, investors should incorporate the risk levels of the assets into the objective function of their portfolio constructions, depending on their risk tolerance.

## 5. Conclusions

Risk diversification for portfolio management is of great interest for both individual and institutional investors. Indeed, various portfolio construction methods have been developed and employed in both individual and industrial trades. Nevertheless, almost all of the portfolio constructions fail to account for the dynamic property of the assets.

To use the dynamic property of the assets, we introduced the method of EOFs into portfolio constructions. The Hilbert transform was used to produce the imaginary part of the analytic signal, from which the complex valued covariance matrix was obtained. The PCA for the covariance matrix enables one to estimate the contribution of each principal axis and to obtain the entropy of the risk contribution distribution. Appropriately constrained optimization methods with respect to the entropy yield the CVRD portfolio construction.

The performance of the CVRD portfolio construction was compared with that of the RP and MRD portfolio constructions. It was confirmed that the annual return of the CVRD portfolio construction outperformed that of the others. In addition, the risk of the assets in the CVRD portfolio construction was well allocated. This result verified that the CVRD portfolio construction succeeded in diversifying the risk of the portfolio.

As the dimensionality reduction of the CVRD portfolio construction, we used the linear PCA. On the other hand, nonlinear dimensionality reductions, such as Isomap and kernel PCA, are expected to provide better risk diversified portfolios. Nevertheless, how to estimate the weight coefficients of the portfolio constructed by the nonlinear dimensionality reductions is a remaining task.

In practice, the time window of estimating the covariance matrix varies on the investors’ policy. Moreover, the accessible test period depends on the resources of the institution where the investor belongs. Hence, the performance of the CVRD portfolio construction seems to vary depending on the size of the time windows and test periods.

Incorporating the nonlinear dimensionality reductions and comprehensive investigation for the effect of the time window and the test period are the focus of our future work. 

## Figures and Tables

**Figure 1 entropy-21-00119-f001:**
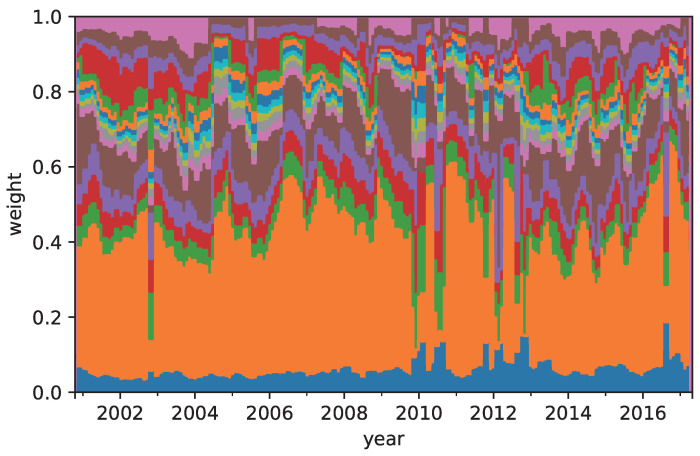
The allocation of the assets in the risk parity (RP) portfolio construction.

**Figure 2 entropy-21-00119-f002:**
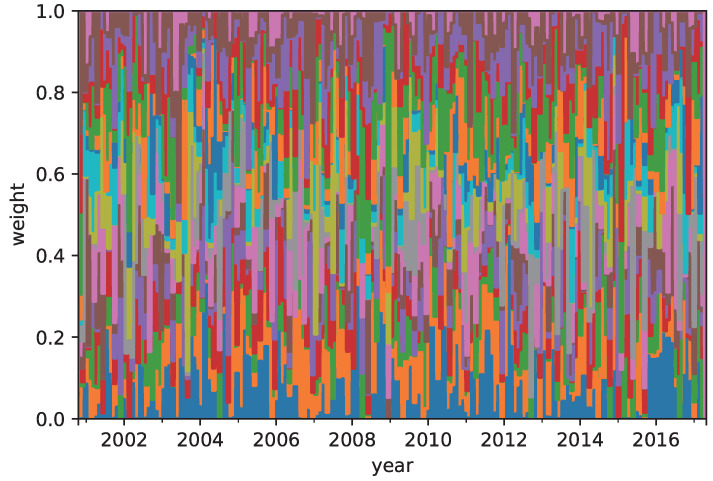
The allocation of the assets in the complex valued risk diversification (CVRD) portfolio construction.

**Figure 3 entropy-21-00119-f003:**
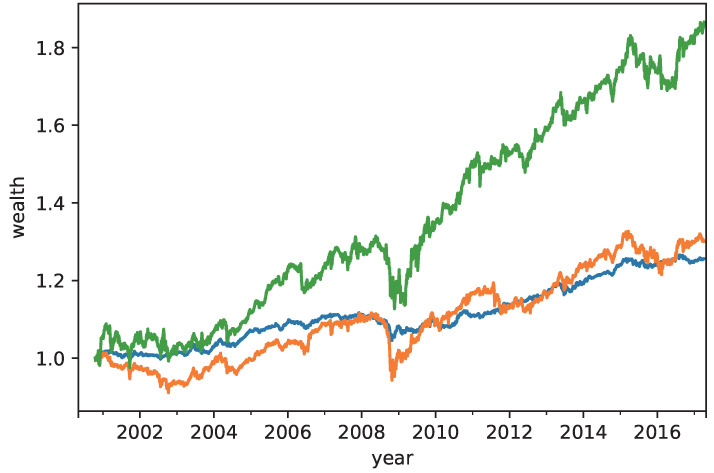
The annual returns of the RP (**blue**), the raximum risk diversification (MRD) (**orange**), and the CVRD (**green**) portfolio constructions.

**Table 1 entropy-21-00119-t001:** Summary of the conventional portfolios. MV: Mean-variance. RP: Risk parity. MRD: Maximum risk diversification.

Portfolio	What is Diversified?	Risk Measure
MV	Asset class	Empirical covariance matrix
RP	Risk	Empirical covariance matrix
MRD	Origin of risk	PCA of empirical covariance matrix

**Table 2 entropy-21-00119-t002:** Asset list.

Name	Type	Definition
TX1	Commodity	10 Year T-Note Futures
XM1	Commodity	Australian 10 Year Bond
CN1	Commodity	Canadian Government 10 Year Note
RX1	Commodity	Eurex Euro Bund
G1	Commodity	Gilt UK
JB1	Commodity	Japan 10 Year Bond Futures
SP1	Index	S&P500
XP1	Index	S&P/ASX 200 (Austraria)
PT1	Index	S&P/TSX 60 Index (Canada)
GX1	Index	DAX (German)
Z1	Index	FTSE100 (UK)
NK1	Index	Nikkei225 (Japan)
AUD	Currency	AUD/USD
CAD	Currency	CAD/USD
EUR	Currency	EUR/USD
GBP	Currency	GBP/USD
JPY	Currency	JPY/USD

**Table 3 entropy-21-00119-t003:** Descriptive statistics of the dataset.

	Mean	Standard Deviation	Skewness	Kurtosis
TY1	0.00661	0.465	−0.284	4.56
XM1	0.000876	0.0621	−0.140	1.81
CN1	0.00935	0.444	−0.222	2.87
RX1	0.01288	0.464	−0.821	6.95
G1	0.00334	0.532	−7.35	214
JB1	0.00414	0.283	−0.571	5.78
SP1	0.210	14.7	−0.250	5.37
XP1	0.624	46.9	−0.289	5.60
PT1	0.0776	7.42	−0.628	7.42
GX1	1.10	90.8	−0.227	4.01
Z1	0.166	61.3	−0.223	3.61
NK1	0.165	194	−0.302	5.44
AUD	0.000037	0.00641	−0.361	6.11
CAD	−0.000026	0.00686	0.0726	2.92
EUR	0.0000400	0.00772	0.0346	1.85
GBP	−0.0000600	0.00939	−0.665	7.55
JPY	0.000695	0.675	−0.147	3.19

**Table 4 entropy-21-00119-t004:** Annual return, risk and, Sharpe ratio of the RP, MRD, and complex valued risk diversification (CVRD) portfolio constructions. Each portfolio is rebalanced every month by previous data for a year.

	RP	MRD	CVRD
Return	1.340	1.621	3.816
Risk	1.728	4.219	6.152
Sharpe Ratio	0.7756	0.384	0.620
Omega Ratio	1.145	1.117	1.205

**Table 5 entropy-21-00119-t005:** Averaged weights of the portfolios.

	RP (%)	MRD (%)	CVRD (%)
TY1	5.37	12.1	7.38
XM1	36.8	17.7	10.7
CN1	5.35	11.4	5.66
RX1	5.80	6.54	5.45
G1	5.33	4.63	6.33
JB1	1.29	4.89	5.37
SP1	1.74	3.48	5.27
XP1	2.19	2.97	5.33
PT1	1.57	3.26	8.42
GX1	1.44	3.96	7.04
Z1	1.79	3.59	4.57
NK1	1.74	3.33	3.29
AUD	2.60	4.50	4.34
CAD	4.17	4.77	5.25
EUR	3.45	4.38	4.92
GBP	3.67	3.56	4.70
JPY	4.13	4.97	5.94

**Table 6 entropy-21-00119-t006:** Averaged weights of the portfolios by type of securities.

	RP [%]	MRD [%]	CVRD [%]
Commodity	71.5	57.3	41.0
Index	10.5	20.6	33.9
Currency	18.0	22.1	25.1

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
