# Peer review of "Complex Valued Risk Diversification"

_entropy, 2019, doi:10.3390/e21020119_

Round 1
Reviewer 1 Report
This paper proposes a portfolio construction method that incorporates the complex valued principal component analysis into the risk diversification portfolio construction. The authors name the proposed method a complex valued risk diversification (CVRD) portfolio construction, and the results show that the CVRD method outperform the conventional risk parity and risk diversification portfolio constructions.
It is an interesting topic, and a new perspective, and CVRD is proposed with entropy. There are some comments and suggestions as follow:
(1) In the Introduction, why is the CVRD method adopted? What is difference between it and traditional models such as the Reference 9? It is suggested that the advantage of the CVRD method be highlighted.
(2) It is suggested that some recent literature be supplemented to highlight the motivation.
(3) In section “2.2 risk parity portfolio”, why is the entropy introduced? It is suggested that the necessity as well as the advantage of the entropy method be emphasized.
(4) In section “2. Related works”, the authors introduce some traditional methods. It is better to make it more concise.
(5) As this paper addressed, the proposed method is verified to outperform the conventional risk parity and risk diversification portfolio constructions. Is there any limitation compared with other methods? It needs more discussion.
Author Response
Response to Reviewer 1 Comments
Dear Reviewer,
We appreciate the time and effort you have dedicated to providing insightful feedback on ways to strengthen our paper. Thus, it is with great pleasure that we resubmit our article for further consideration. We have incorporated changes that reflect the detailed suggestions you have graciously provided. We also hope that our edits and the responses we provide below satisfactorily address all the issues and concerns you have noted.
To facilitate your review of our revisions, the following is a point-by-point response to the questions and comments delivered in your comments.
Point 1: In the Introduction, why is the CVRD method adopted? What is difference between it and traditional models such as the Reference 9? It is suggested that the advantage of the CVRD method be highlighted.
Response 1: The CVRD can incorporate the dynamic properties of the returns of assets into portfolios by using the analytical signal generated by the Hilbert transform. This is the dominance of applying the EOFs to the portfolio construction. This advantage of the EOFs and the CVRD is added in paragraph 4 & 5 in Introduction with reference of an application of the Hilbert transform, Kurbatsky et al. (2014).
Point 2: It is suggested that some recent literature be supplemented to highlight the motivation.
Response 2: According to your recommendation, we added recent literature of the risk diversification in paragraph 3 in Introduction with reference 17, Bernardi et al. (2018).
Point 3: In section “2.2 risk parity portfolio”, why is the entropy introduced? It is suggested that the necessity as well as the advantage of the entropy method be emphasized.
Response 3: The key concept of the MRD and CVRD is diversifying risk contributions of asset allocations. To do so, the entropy of the probability distribution of the risk contributions is used as an objective function because the principle of maximum entropy realizes diversified states. This explanation is inserted in paragraph 2 in subsection 1.3 “Risk diversification”.
Point 4: In section “2. Related works”, the authors introduce some traditional methods. It is better to make it more concise.
Response 4: To provide a concise explanation of the traditional method, we introduce table 1 and short sentences in the beginning of section 1 “Related works”.
Point 5: As this paper addressed, the proposed method is verified to outperform the conventional risk parity and risk diversification portfolio constructions. Is there any limitation compared with other methods? It needs more discussion.
Response 5: As well as the MRD, the rebalancing frequency of the CVRD tends to be higher than that of others. This might cause more transaction costs. Also, the CVRD does not consider risk level of assets. Hence, one should incorporate the risk level into the objective function depending on their risk tolerance if necessary. This discussion is added as paragraph 4 in subsection 3.2 “Performance test of portfolios”.
Again, thank you for giving us the opportunity to strengthen our manuscript with your valuable comments and queries. We have worked hard to incorporate your feedback and hope that these revisions persuade you to accept our submission.
Sincerely,
Yusuke Uchiyama, Takanori Kadoya
MAZIN Inc.,
Kei Nakagawa
Nomura Asset Management Co., Ltd.

Reviewer 2 Report
The Hilbert transform is in the core of many time series forecasting and risk analysis methods, see V. Kurbatsky et al Forecasting nonstationary time series based on Hilbert-Huang transform and machine learning, Springer, DOI:10.1134/S0005117914050105. This should be outlined in the introduction. Some typos, ref. e.g. line 133 and others must be fixed as well.
The main flaw of this paper is usage of linear PCA. Instead the nonlinear dimensionality reduction using Isomap should be employed or at least it must be discussed.
Author Response
Response to Reviewer 2 Comments
Dear Reviewer,
We appreciate the time and effort you have dedicated to providing insightful feedback on ways to strengthen our paper. Thus, it is with great pleasure that we resubmit our article for further consideration. We have incorporated changes that reflect the detailed suggestions you have graciously provided. We also hope that our edits and the responses we provide below satisfactorily address all the issues and concerns you have noted.
To facilitate your review of our revisions, the following is a point-by-point response to the questions and comments delivered in your comments.
Point 1: The Hilbert transform is in the core of many time series forecasting and risk analysis methods, see V. Kurbatsky et al Forecasting nonstationary time series based on Hilbert-Huang transform and machine learning, Springer, DOI:10.1134/S0005117914050105. This should be outlined in the introduction.
Response 1: According to your recommendation, in paragraph 5 in Introduction, we refer an effectiveness of the Hilbert transform to capture the dynamic properties of signals related to the Hilbert-Huang transform in Kurbatsky (2014).
Point 2: Some typos, ref. e.g. line 133 and others must be fixed as well.
Response 2: We fixed typos including you pointed out again and again in the recent version.
Point 3: The main flaw of this paper is usage of linear PCA. Instead the nonlinear dimensionality reduction using Isomap should be employed or at least it must be discussed
Response 3: As you pointed out, nonlinear dimensionality reductions are expected to show better performance than linear ones. On the other hand, the aim of our research is getting the weight coefficients of the portfolio. If we employ the nonlinear dimensionality reductions, it is quite a difficult task to obtain the weight coefficients of the portfolio because of the nonlinearity of mapping. To obtain the weight coefficients of the portfolio directly, we used linear PCA. It is, however, a challenging task to reconstruct the weight coefficients of the portfolio from the nonlinear mapping. That should be one of our future works. These explanations are reflected in third paragraph in section 2 “Complex valued risk diversification portfolio construction” and paragraph 4 & 6 in section 4 “Conclusion”.
Again, thank you for giving us the opportunity to strengthen our manuscript with your valuable comments and queries. We have worked hard to incorporate your feedback and hope that these revisions persuade you to accept our submission.
Sincerely,
Yusuke Uchiyama, Takanori Kadoya
MAZIN Inc.,
Kei Nakagawa
Nomura Asset Management Co., Ltd.

Reviewer 3 Report
Please see the attached file.

Author Response
Response to Reviewer 3 Comments
Dear Reviewer,
We appreciate the time and effort you have dedicated to providing insightful feedback on ways to strengthen our paper. Thus, it is with great pleasure that we resubmit our article for further consideration. We have incorporated changes that reflect the detailed suggestions you have graciously provided. We also hope that our edits and the responses we provide below satisfactorily address all the issues and concerns you have noted.
To facilitate your review of our revisions, the following is a point-by-point response to the questions and comments delivered in your comments.
Point 1: The main concern in their proposed method is whether their proposed method overestimate the real counterpart. It is well known that the traditional portfolio estimation is overestimated, see, for example, Bai, et al. (2009) and the references therein for more information. The authors should address this issue and make sure there is not serious overestimation.
Response 1: Based on the correction formula in Bai, et al. (2009), we estimate the correction coefficient in our case. As a result, the coefficient is obtained as 1.034. We think this value does not affect the results of our analysis, but refer this formula with related papers in the paragraph 1 in the subsection 3.2 “Performance test of portfolios”.
Point 2: Another concern is that risk and Sharpe ratio are not good measures to measure risk while Omega ratio performs better (Guo, et al., 2017). The authors should include Omega ratio in their analysis.
Response 2: As is pointed out in Guo et al. (2017) and others, omega ratio is a better performance measure because it can include all of the information about statistical moments. Thus, we added the omega ratios of the portfolios in our study and compare them in the table 4. In fact, the omega ratios also verify our result that the CVRD method outperforms others.
Point 3: The authors should have a data section to describe data clearly.
Point 4: The authors should explain how they get their results.
Response3& 4: To describe the data clearly, we add subsection 3.1 “Data description” with tables 2 and 3. Here, the asset types and their statistics are shown. In addition, subsection 3.2 “Performance test of portfolios” explains how to get our results and why the CVRD outperforms others.
Point 5: The authors claim that the proposed method is verified to outperform the conventional risk parity and risk diversification portfolio constructions. They should explain why their proposed method outperform the conventional risk parity and risk diversification portfolio constructions. Also, the conventional risk parity and risk diversification portfolio constructions may perform poorly. The authors should compare some portfolio constructions that have been demonstrated to perform well in the comparison. For example, the approach developed by Li, et al. (2018).
Response 5: According to the approach that the review recommended, we compare the values of the weight coefficients of the portfolios. As a result, it is seen that the CVRD holds larger amounts of indexes than the RP and MRD while the RD is biased on the commodities. In addition, the weight coefficients of the CVRD are the most diversified. That means the CVRD resolves the biased balance of the RP and MRD portfolio constructions. In other words, the CVRD can diversify the origins of the risks of the assets. This explanation appears in subsection 3.2.
Again, thank you for giving us the opportunity to strengthen our manuscript with your valuable comments and queries. We have worked hard to incorporate your feedback and hope that these revisions persuade you to accept our submission.
Sincerely,
Yusuke Uchiyama, Takanori Kadoya
MAZIN Inc.,
Kei Nakagawa
Nomura Asset Management Co., Ltd.

Round 2
Reviewer 1 Report
Thanks the efforts paid on revision. I recommend to accept it.
Reviewer 3 Report
The revision is in good quality and I am satisfied with the revision. Just the authors should check the references carefully. There are some mistakes in the references, e.g.
Z. Bai, and H. Liu, Enhancement of the applicability of Markowitz’s portfolio optimization by utilizing random matrix theory, Math. Finan. 2009, 19,639-–667.
Should be
Z. Bai, H. Liu, and W.K. Wong, Enhancement of the applicability of Markowitz’s portfolio optimization by utilizing random matrix theory, Math. Finan. 2009, 19,639-–667.